# CUPID: POSE-GROUNDED GENERATIVE 3D RECONSTRUCTION FROM A SINGLE IMAGE

## ABSTRACT

This work proposes a new generation-based 3D reconstruction method, named CUPID, that accurately infers the camera pose, 3D shape, and texture of an object from a single 2D image. CUPID casts 3D reconstruction as a conditional sampling process from a learned distribution of 3D objects, and it jointly generates voxels and pixel-voxel correspondences, enabling robust pose and shape estimation under a unified generative framework. By representing both input camera poses and 3D shape as a distribution in a shared 3D latent space, CUPID adopts a two-stage flow matching pipeline: (1) a coarse stage that produces initial 3D geometry with associated 2D projections for pose recovery; and (2) a refinement stage that integrates pose-aligned image features to enhance structural fidelity and appearance details. Extensive experiments demonstrate CUPID outperforms leading 3D reconstruction methods with an over 3 dB PSNR gain and an over 10% Chamfer Distance reduction, while matching monocular estimators on pose accuracy and delivering superior visual fidelity over baseline 3D generative models.

## 1 INTRODUCTION

When we look at an object, we instinctively recognize the viewpoint from which it is seen—whether from the front, side, or back. This ability suggests that, based on a single view, humans can "reconstruct" a canonical 3D model of the object, including its shape and texture, and also estimate this viewpoint relative to the model (Sekuler & Palmer, 1992). Accurately determining both a canonical model of the object and an object-centric pose of the viewpoint is essential for effective interaction with the object (Gibson, 2014). For example, when we reach to grasp a cup, we can easily adjust the position of our hand based on our estimated viewpoint in relation to the cup's canonical orientation. Therefore, it is crucial for embodied AI that robots develop similar abilities: from any given view, they should be able to i) recover the 3D model of an object in its canonical pose (often referred to as the "prototypical" exemplar, which has a privileged status) and ii) estimate the current camera viewpoint relative to the canonical object frame[1]. We call the task of estimating from images both an object's canonical 3D model and object-centric camera poses as *pose-grounded 3D reconstruction*. Figure 1 illustrates the basic ideas with exemplar results produced by our new method.

Most existing learning-based 3D reconstruction or generation works only focus on one aspect of pose-grounded 3D reconstruction: either estimating relative poses between multiple input images (Wang et al., 2024; Zhang et al., 2024a), or reconstructing a view-centric 3D model without recovering the canonical model (Schönberger & Frahm, 2016; Hong et al., 2023), or generating 3D models without estimating the object-centric camera pose of the input image(s) (Xiang et al., 2025). To the best of our knowledge, no work has adequately addressed the two related objectives in the pose-grounded 3D reconstruction task within a unified framework.

Specifically, most multi-view based 3D reconstruction methods focus on recovering a 3D model for the observed parts from image(s) by leveraging correlations between images and view-centric depths (Yang et al., 2024), point clouds (Wang et al., 2025a;c), or radiance fields (Hong et al., 2023). When the poses of the input images are not provided, however, separate PnP or deep learning-based methods are often adopted for relative pose estimation (Wang et al., 2023; Xu et al., 2024c; Wang et al., 2024; Chen et al., 2022). Furthermore, all these methods approach reconstruction as a 2D-to-3D inverse problem based on the provided image(s), failing to account for the uncertainty caused by

---

[1]To simplify the presentation, "object" refers to the object model in its canonical pose, and "camera pose" refers to the object-centric camera pose in the following sections.

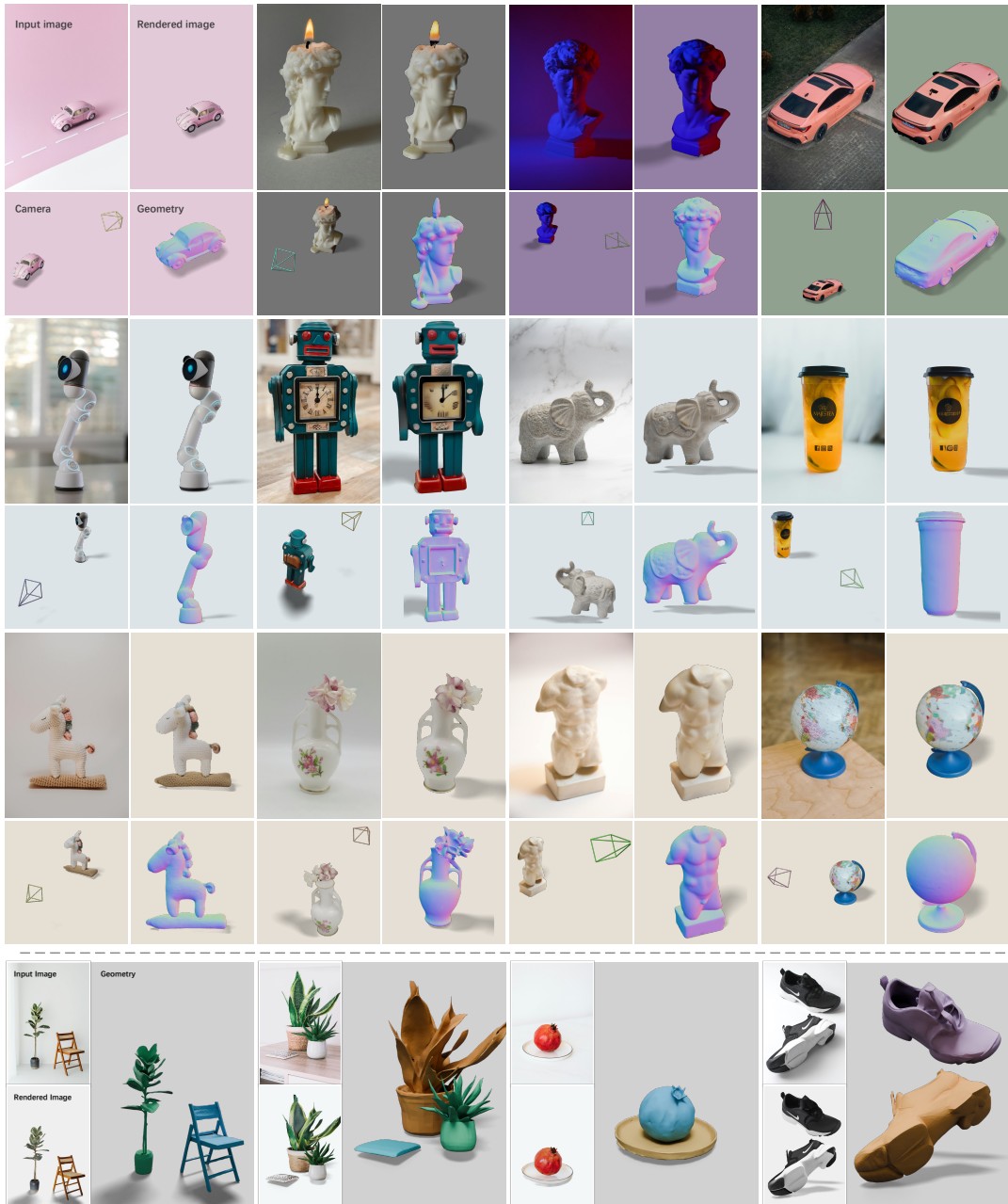

Figure 1: **Results for pose-grounded *generative 3D reconstruction* from a single test image.** Given an input image (top left), CUPID estimates camera pose (bottom left) and reconstructs 3D model (bottom right), re-rendering the input (top right). It is robust to changes in scale, placement, and lighting while preserving fine texture, and supports component-aligned scene reconstruction (bottom row). All results are produced in seconds via feed-forward sampling of the learned model.

self-occluded or unseen regions. As a result, they struggle to accurately recover complete 3D objects or scenes from limited views. Ultimately, none of these reconstruction-based methods produces a canonical 3D model or estimates the viewpoint of each image relative to the canonical object pose, as desired for embodied AI applications.

Conversely, generative methods can synthesize canonically posed 3D objects (Lu et al., 2025) from a single image, offering priors over textures (Zhang et al., 2024b), parts (Yang et al., 2025b) and actionable labels (Liu et al., 2024a) that are difficult to obtain from reconstruction alone. Further, current image-conditioned 3D generators typically infer plausible variations rather than produce a

faithfully reconstructed 3D model based on the given image(s). As a result, they can exhibit shape inconsistencies and color shifts when real inputs deviate from the synthetic training distribution (Xiang et al., 2025). Moreover, because pose is not explicitly modeled, integrating the so-generated results into 3D reconstruction typically requires costly pose optimization, and alignment remains fragile when the generated 3D object deviates from the observation (Yao et al., 2025; Dong et al., 2025). Thus, single-image (or few-image based) object reconstruction from a learned generative 3D prior remains largely an open challenge.

In this work, we address this challenge by identifying *pose grounding* as the key gap between 3D generation and reconstruction: to explain a single image, a model must recover dense 3D–2D correspondences between the object and the input viewpoint. To marry the benefits of reconstruction and generation, we introduce CUPID (for "in-Cube Pixel Distribution"), a pose-grounded generative framework that jointly models 3D voxels and 2D pixels within a unified framework. CUPID adopts a sparse voxel representation and a two-stage training scheme inspired by Xiang et al. (2025). First, a flow model jointly samples 3D points and their 2D pixel correspondences, enabling efficient pose recovery via PnP (Abdel-Aziz et al., 2015). Second, conditioned on the recovered pose, CUPID refines geometry and appearance based on reprojection consistency. Unlike image-conditioned 3D generators that rely on global image features, CUPID back-projects pose-aligned locally-attended pixel features into the voxel grid, thereby improving geometric alignment and appearance fidelity. In essence, our method conducts Bayesian inference for the 3D reconstruction by leveraging the learned 3D generative distribution conditioned on the pose and visual cues of the input image, hence significantly closes the gap between generation and reconstruction. As Figure 1 shows, this combination of generative priors and observations allows CUPID to produce high-fidelity 3D reconstructions that are both geometrically accurate and texturally faithful to the input image.

The primary contributions of this work can be summarized as below:

- A unified generative framework, CUPID, that grounds 3D synthesis in a single 2D image. For the first time, we jointly model 3D object and dense 2D-to-3D correspondences, ensuring the generated 3D object is explicitly tied to the input view.

- A novel pose-conditioned refinement strategy. Our two-stage approach decouples pose estimation from the generation of detailed geometry and texture, a mechanism critical for achieving high-fidelity reconstructions with strong geometric and textural consistency.

- State-of-the-art results across both reconstruction and pose estimation. CUPID sets a new standard for single-image reconstruction, improving fidelity by over 3 dB PSNR against top 3D reconstruction models, while achieving pose accuracy competitive with or superior to point-map estimators.

## 2 METHOD

### 2.1 PROBLEM DEFINITION AND METHOD OVERVIEW

We formulate generative reconstruction as estimating the joint posterior $p(\mathcal{O}, \boldsymbol{\theta} \mid \mathbf{I}^{\text{cond}})$ under the observation constraint $\mathbf{I}^{\text{cond}} = \mathcal{P}(\mathcal{O}, \boldsymbol{\theta})$, where $\mathbf{I}^{\text{cond}}$ is the input image, $\mathcal{O}$ is the 3D object, $\boldsymbol{\theta}$ is the object-centric camera pose, and $\mathcal{P}(\cdot, \cdot)$ projects the 3D object to the image.

We first use an encoder $\varphi$ to map the 3D object and camera pose to a volumetric latent feature $\mathbf{z} = \varphi(\mathcal{O}, \boldsymbol{\theta})$, which can be decoded into 3D representations (e.g., Gaussian splats or meshes) by specific decoders (Shen et al., 2023; Kerbl et al., 2023). We use Rectified Flow (Lipman et al., 2022) for the latent $\mathbf{z}$ generation. The model interpolates between data $\mathbf{z}_0$ and noise $\boldsymbol{\epsilon}$ over time $t$ via $\mathbf{z}_t = (1-t)\mathbf{z}_0 + t\boldsymbol{\epsilon}$. The reverse process follows a time-dependent velocity field $\mathbf{v}(\mathbf{z}_t, \mathbf{I}^{\text{cond}}, t) = \nabla_t \mathbf{z}_t$ that drives noisy samples toward the data distribution. We parameterize $\mathbf{v}$ with a neural network $\mathbf{v}_\phi$ and train it using the Conditional Flow Matching (CFM) objective:

$$\mathcal{L}_{\text{CFM}}(\phi) = \mathbb{E}_{t, \mathbf{z}_0, \boldsymbol{\epsilon}} \left\| \mathbf{v}_\phi(\mathbf{z}_t, \mathbf{I}^{\text{cond}}, t) - (\boldsymbol{\epsilon} - \mathbf{z}_0) \right\|_2^2. \tag{1}$$

We will first describe the representation of object and camera pose in Section 2.2. To enable efficient 3D generation, we utilize a cascaded flow modeling approach for $\mathbf{v}_\phi$ (Xiang et al., 2025). In the first stage, we generate an occupancy cube along with the camera pose. The second stage predicts the 3D shape and texture features in the occupied regions based on the outputs from the first stage. This process will be detailed in Section 2.3. The pipeline is illustrated in Figure 2.

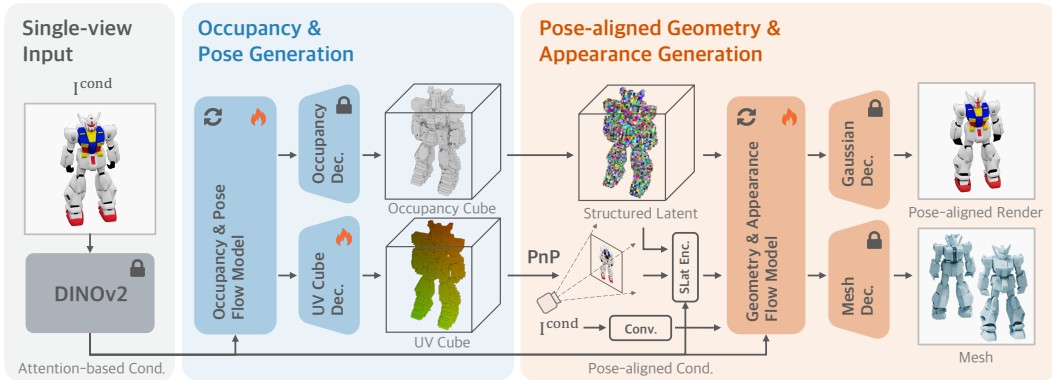

Figure 2: **Overview of CUPID**. From a single input image, CUPID first generates an occupancy and a UV cube in canonical space. A *Perspective-n-Point (PnP)* solver (*i.e.*, Equation 2) then recovers the camera pose. Using this recovered camera pose, we extract pose-aligned conditioning latents and visual features, along with noisy structured latents, to generate the geometry and appearance features, which will be decoded to the 3D representation and mesh.

## 2.2 REPRESENTATION OF 3D OBJECT AND CAMERA POSE

**3D object representation.** To enable network training, we first tensorize 3D objects with a voxel-based representation: $\mathcal{O} \triangleq \{\mathbf{x}_i, \mathbf{f}_i\}_{i=1}^{L}$, where $\mathbf{x}_i \in \mathbb{R}^3$ represents the coordinates of the $i$-th active voxel in a cube, and $\mathbf{f}_i$ encodes associated features derived from multi-view DINOv2 (Oquab et al., 2023) feature aggregation, subsequently compressed by a 3D variational autoencoder encoder (VAE). Here, $L$ denotes the number of voxels. This voxel set can be decoded back to 3D formats such as mesh or Gaussian splat (Xiang et al., 2025).

**Camera pose parameterization.** Camera pose $\boldsymbol{\theta}$ is represented by the projection matrix $\mathbf{P} = \mathbf{K}[\mathbf{R}|\mathbf{t}] \in \mathbb{R}^{3\times 4}$, where $\mathbf{K}$ is intrinsic, and $(\mathbf{R}, \mathbf{t})$ are extrinsic parameters mapping object to camera space. Inspired by recent pose estimation work (Wang et al., 2024), we propose a dense 3D pose representation[2] via an in-cube pixel distribution (CUPID). Specifically, we reparameterize $\boldsymbol{\theta}$ as dense 3D-2D correspondence: $\boldsymbol{\theta} \triangleq \{\mathbf{x}_i, \mathbf{u}_i\}_{i=1}^{L}$ where $\mathbf{u}_i : (u_i, v_i) \in [0, 1]^2$ is the normalized 2D pixel coordinates anchored at the 3D voxel $\mathbf{x}_i \in \mathcal{O}$. Each anchor's pixel coordinate is obtained by $\mathbf{u}_i = \pi(\mathbf{P}, \mathbf{x}_i)$, where $\pi$ denotes perspective projection. Such reparameterized pose, i.e., 3D-2D correspondence can be presented as 3D UV cube. Given these correspondences, we recover the global camera matrix $\mathbf{P}^*$ using a least-squares solver (Abdel-Aziz et al., 2015):

$$\mathbf{P}^* = \arg\min_{\mathbf{P}} \sum_{i=1}^{L} \left\| \pi(\mathbf{P}, \mathbf{x}_i) - \mathbf{u}_i \right\|^2. \tag{2}$$

We then decompose $\mathbf{P}^*$ into $(\mathbf{K}, \mathbf{R}, \mathbf{t})$ via RQ decomposition. Unlike image-based pose representations (e.g., 2D ray or point maps (Zhang et al., 2024a; Wang et al., 2024)), we reparameterize pose within a 3D cube. Intuitively, the $(u, v)$ coordinates act like view-dependent colors defined on a 3D occupancy grid (see the UV cube in the Figure 2). Under this view, joint object–pose generation transforms to producing a 3D object with view-dependent colors.

## 2.3 CASCADED FLOW MODELING

We employ a two-stage cascaded flow model to jointly sample a 3D object and its corresponding camera pose, *i.e.*, $\mathbf{z} = \{(\mathbf{x}_i, \mathbf{f}_i), (\mathbf{x}_i, \mathbf{u}_i)\}_{i=1}^{L}$. In the first stage, the flow model for occupancy and pose generation ($\mathcal{G}_{\mathbf{S}}$) generates (i) an occupancy cube that indicates active voxels and (ii) a UV cube that contains object-centric camera pose information (Section 2.2). In the second stage, conditioned on the predicted occupancy and camera pose, the pose-aligned geometry appearance generation model $\mathcal{G}_{\mathbf{L}}$ synthesizes DINO features $\mathbf{f}_i$ for each active voxel, yielding the final $\mathbf{z}$.

---

[2]While one can use a 12D vector for pose, such low-dimension parameterization can hinder neural optimization (Zhang et al., 2024a).

**Occupancy and pose generation.** In this stage, given a conditioning image $\mathbf{I}^{\text{cond}}$, we generate an occupancy cube and a UV cube with resolution $r$. The occupancy cube $\mathbf{G}_o \in \{0,1\}^{r \times r \times r \times 1}$ encodes binary values indicating active/nonactive voxels. The UV cube $\mathbf{G}_{uv} \in [0,1]^{r \times r \times r \times 2}$ stores normalized pixel coordinates $(u, v)$ for each voxel. Note that voxels sharing the same pixel coordinates are in the same ray[3]. Next, we train a 3D VAE to compress the UV cube into a low resolution feature grid $\mathbf{S}_{uv} \in \mathbb{R}^{s \times s \times s \times C}$ to improve computational efficiency. For fine-tuning TRELLIS, we concatenate the original feature grid $\mathbf{S}_o$ with $\mathbf{S}_{uv}$ and incorporate linear layers at both the input and output of the flow network $\mathcal{G}_{\mathbf{S}}$. Once the occupancy cube and the UV cube are generated, we can obtain $\{(\mathbf{x}_i, \mathbf{u}_i(\boldsymbol{\theta}))\}_{i=1}^L$ by collecting active voxels and solving the camera pose with Equation 2.

**Pose-aligned geometry and appearance generation.** In this stage, we aim to generate the detailed 3D latent $\{\mathbf{f}_i\}_{i=1}^L$ only at active voxels. Our experiments show that the original $\mathcal{G}_{\mathbf{L}}$ from TRELLIS, which uses globally-attended image information, often suffers from color drift and loss of fine details (see Figure 8 in Appendix A.4). To address this issue, we leverage the calculated camera pose from the first stage to inject locally-attended pixel information in each voxel.

Specifically, we incorporate a single-view, pixel-aligned 3D latent derived from the input image and our generated camera pose to calculate the features located at the $i$-th voxel as follows:

$$\mathbf{f}_i^{\text{DINO}} = \text{BilinearInterp}(\mathbf{u}_i, \text{DINO}(\mathbf{I}^{\text{cond}})) \in \mathbb{R}^{1024},$$
$$\{\mathbf{f}_i^{\text{h}}\}_{i=1}^L = \text{SlatEncoder}\left(\{\mathbf{x}_i, \mathbf{f}_i^{\text{DINO}}\}_{i=1}^L\right), \quad \mathbf{f}_i^{\text{h}} \in \mathbb{R}^8, \tag{3}$$

where each voxel's pixel coordinate $\mathbf{u}_i$ is obtained by projecting the coordinates of the $i$-th 3D voxel center onto the image plane with the calculated camera pose, $\text{BilinearInterp}$ denotes bilinear interpolation and $\text{SlatEncoder}$ is the 3D VAE encoder. While DINO captures high-level semantics, it loses low-level cues needed for precise 3D geometry and appearance reconstruction. To compensate for it, we extract complementary low-level features $\mathbf{f}^{\text{l}}$ from $\mathbf{I}^{\text{cond}}$ with a lightweight convolutional head and sample them at $\mathbf{u}_i$ via $\text{BilinearInterp}$. Finally, for each voxel, we concatenate the current noisy voxel feature $\mathbf{f}_i^t$ at time step $t$ with the pixel-aligned features and fuse them into the flow transformer with a linear layer: $l_t = \text{Linear}\left([\mathbf{f}_i^t \oplus \mathbf{f}_i^{\text{h}} \oplus \mathbf{f}_i^{\text{l}}]\right)$. This pose-aligned fusion significantly improves 3D geometry accuracy and appearance fidelity with respect to the input image.

## 3 COMPONENT-ALIGNED SCENE RECONSTRUCTION

Our pose-grounded generative framework seamlessly extends to scene-level compositional reconstruction by leveraging explicit 3D-to-2D correspondences for object-centric to view-centric transformations. We harness foundation models for object segmentation (Ravi et al., 2024), generate each 3D object, and compose them into a scene with a global depth prior (Wang et al., 2025d). Figure 3 shows one such result with our method. More results are given in Figure 9 of Appendix A.3.

**Occlusion-aware 3D generation.** Images with multiple objects often contain mutual occlusions, which our synthetic training data do not model. To address this, we introduce random masks on the conditioning image during fine-tuning, inspired by Amodal3R (Wu et al., 2025). We adapt their masking strategy to our architecture; see Appendix A.2 for implementation. Then, given an input image with multiple objects and their corresponding object masks, we reconstruct each object independently with our occlusion-aware generator, yielding dense 3D–2D correspondences per object.

**Multi-component composition.** Since absolute depth is not preserved across objects, direct 3D-2D alignment to the image is ill-posed. We address this using MoGe (Wang et al., 2025c) to predict a global pointmap, thereby reducing 3D–2D alignment to 3D–3D alignment. For the $k$-th object, we collect the matched pair $(\mathbf{x}_i^k, \mathbf{u}_i)$, where $\mathbf{x}_i^k$ is the coordinates of a visible 3D point on the generated 3D object and $\mathbf{u}_i$ is the corresponding pixel coordinates within the mask $\mathbf{M}^k$. We query the MoGe pointmap to obtain $\mathbf{p}_i = P(\mathbf{u}_i)$ in the camera frame, then estimate a per-object similarity transformation $\mathcal{S}_k = (s_k, \mathbf{R}_k, \mathbf{t}_k)$ via the Umeyama method (Umeyama, 2002) on pairs $(\mathbf{x}_i^k, \mathbf{p}_i)$. Applying $\mathcal{S}_k$ places each object in the shared camera frame, yielding a component-aligned scene reconstruction. This composition can extend to multi-view input using VGGT (Wang et al., 2025a), allowing for more flexible input and transforming partial geometry into holistic geometry.

---

[3]We do not enforce occlusion as recent studies suggest that transformers can effectively model light transport (Jin et al., 2024; Zeng et al., 2025)

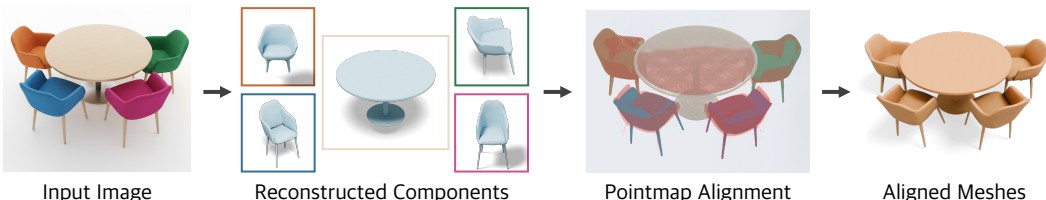

Input Image      Reconstructed Components      Pointmap Alignment      Aligned Meshes

Figure 3: **Component-aligned scene reconstruction.** For a scene with multiple objects, our method can rebuild each object using the occlusion-aware 3D generator and then solve 3D–3D similarity transformation to accurately recompose the scene.

## 4 EXPERIMENTS

We evaluate the most relevant baselines on three tasks: monocular geometry prediction, input-view consistency, and single-image-to-3D reconstruction quality in Section 4.1. We do not include per-scene optimization methods, focusing on learning-based systems. We also ablate different pose-conditioning designs in Section 4.2. Implementation details are in Appendix A.1. For 3D generation methods that lack camera poses, we provide qualitative comparisons in the Appendix A.4.

### 4.1 EVALUATION

**Baselines.** We compare against three complementary families for single image 3D reconstruction: point-map regression, view-centric 3D reconstruction, and 3D generation with post-hoc pose alignment. Each probes a distinct capability: visibility-limited lifting, view-centric full 3D recovery, and decoupled canonical 3D object generation and pose estimation.

*Point-map regression.* VGGT (Wang et al., 2025a) and MoGe (Wang et al., 2025c) simultaneously predict per-pixel 3D points and view-centric camera pose for 2D-to-3D reconstruction. These methods reconstruct only the geometry visible in the input view, often lacking robust priors for occluded regions. *View-centric 3D reconstruction.* LRM (Hong et al., 2023) and LaRa (Chen et al., 2024) generate view-centric 3D models, eliminating the need to estimate camera parameters at test time. LRM directly regresses 3D objects in view space, while LaRa enhances the input with novel views from Zero123++ (Shi et al., 2023) before reconstruction. We adopt an open-source implementation of LRM (He & Wang, 2023) for comparisons. *3D generation with post-hoc alignment.* OnePoseGen (Geng et al., 2025) integrates a learned 3D generator (Ye et al., 2025), which produces canonical 3D models, with a robust object pose estimator (Wen et al., 2024) to align the 3D model with the input view. This approach evaluates whether decoupling 3D generation from pose estimation and applying post-hoc alignment can achieve reprojection consistency.

**Evaluation.** We evaluate three aspects: (i) monocular geometry accuracy (following (Wang et al., 2025c)), (ii) input-view consistency to assess reprojection alignment and appearance fidelity, and (iii) novel-view consistency to assess full 3D fidelity. Experiments are conducted on Toys4K (Sto-janov et al., 2021) and GSO (Downs et al., 2022), containing approximately 3K synthetic and 1K real-world objects, respectively. For novel views, we render four uniform views rotating around the vertical axis. Note that tasks (i) and (ii) are particularly challenging for generative approaches, as they require producing an object-centric camera pose for view-centric evaluation.

**Monocular geometry accuracy.** Monocular geometry evaluates visible geometry in the input image. We construct ground-truth point clouds by unprojecting depth maps with camera parameters. We report Mask IoU, Chamfer Distance (CD), and F-score to assess point-cloud quality, providing both mean and median statistics to mitigate outlier sensitivity in generative methods. Since VGGT outputs point maps without object masks, we use ground-truth masks to avoid background contamination. For full 3D methods, we render and unproject the depth map to get the visible point cloud, and use rendered alphas as the object mask. Predicted point clouds are aligned to ground truth using the scale–shift alignment from MoGe (Wang et al., 2025c). Results are in Table 1.

Our approach significantly outperforms full 3D reconstruction methods across all metrics. Notably, it reduces the average CD from LRM by approximately 50% on the GSO dataset, with even greater improvements in median statistics. These results highlight the superiority of our canonical object-centric reconstruction compared to the view-centric approaches of LRM and LaRa. OnePoseGen falls short on Toys4K and GSO because these datasets contain geometric symmetries and textureless

Table 1: **Monocular geometry accuracy.** CUPID outperforms all 3D reconstruction and generation baselines and matches point-map regression methods that predict only partial geometry. VGGT uses a ground-truth object mask, which may overestimate accuracy.

| Method | 3D | Toys4k | | | | | GSO | | | | |
| | | mIOU (avg)↑ | CD (avg)↓ | CD (med)↓ | F-score (0.01)↑ | F-score (0.05)↑ | mIOU (avg)↑ | CD (avg)↓ | CD (med)↓ | F-score (0.01)↑ | F-score (0.05)↑ |
|---|---|---|---|---|---|---|---|---|---|---|---|
| VGGT | × | – | 1.144 | 0.498 | 61.85 | 95.90 | – | 1.396 | 0.388 | 65.98 | 95.95 |
| MoGe | × | 92.80 | 1.284 | 0.581 | 58.54 | 95.31 | 96.18 | 1.743 | 0.575 | 58.99 | 94.68 |
| OnePoseGen | ✓ | 9.34 | 153.2 | 59.92 | 6.11 | 24.10 | 12.16 | 116.2 | 60.56 | 7.28 | 25.77 |
| LaRa | ✓ | 68.11 | 32.15 | 16.59 | 18.57 | 57.67 | 70.63 | 34.23 | 19.36 | 13.48 | 49.95 |
| OpenLRM | ✓ | 86.26 | 2.726 | 1.291 | 40.42 | 90.60 | 91.35 | 3.741 | 1.858 | 34.14 | 87.20 |
| Ours | ✓ | **92.43** | **2.534** | **0.236** | **69.82** | **97.76** | **95.27** | **1.823** | 0.434 | **61.01** | **95.59** |

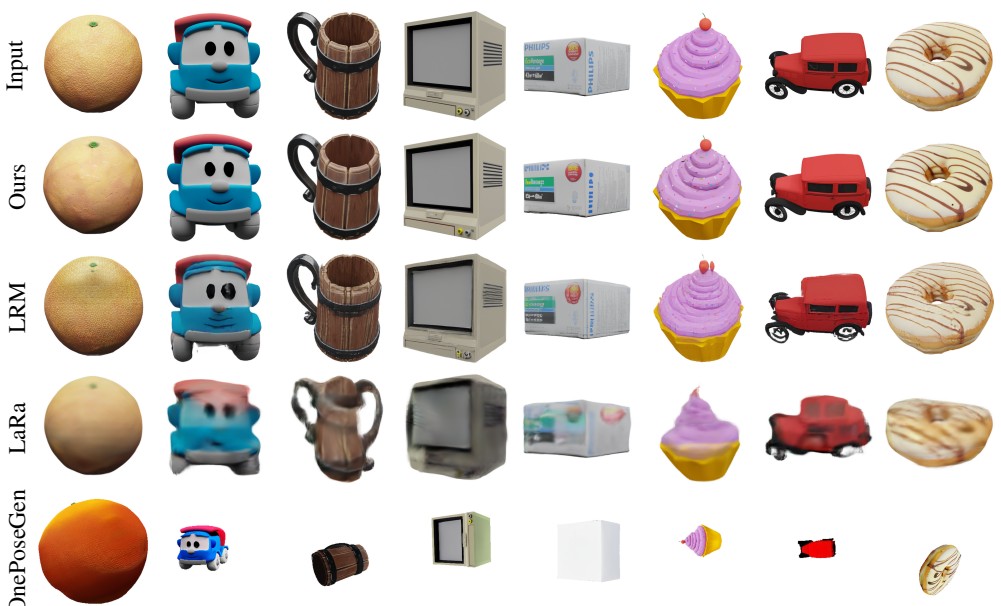

Figure 4: **Comparison of *generative reconstruction* from a single image.** Our method produces the highest-fidelity geometry and appearance; LRM hallucinates incorrect details, LaRa is overly blurry due to 2D diffusion inconsistencies, and OnePoseGen frequently fails to register pose reliably.

regions that cause its registration modules to fail. Our method is also highly competitive with VGGT and MoGe across all metrics, though these methods only produce partial geometries.

**Input view consistency.** We evaluate the input view consistency to measure the alignment between the 3D reconstruction and the input image. We quantify the difference between the re-rendered image and the input image, reporting PSNR, SSIM, and LPIPS in Table 2. We also provide qualitative comparisons in Figure 4. Our method achieves substantial PSNR improvements over existing reconstruction-based methods: 36% and 44% over LaRa on Toys4K and GSO respectively, and 13% and 11% over LRM on the same datasets. We find that LaRa's texture quality is affected by inconsistencies in the generated novel views, while LRM struggles to predict detailed textures. OnePoseGen can produce plausible 3D objects but suffers severely from texture misalignment (e.g., color shifting) and imperfect pose estimation. Our method achieves strong pose accuracy while retaining high-quality texture details, as evidenced by the zoomed-in views in Figure 4.

**Full 3D evaluation.** Evaluating single-image-to-3D is challenging because many distinct 3D shapes can explain a single view. We therefore emphasize (1) full-3D qualitative assessments (Figure 4) and (2) quantitative semantic evaluations using CLIP (Radford et al., 2021) (Table 3). Our method consistently outperforms all baselines. Moreover, it reconstructs high-quality 3D objects and scenes from images in the wild (Figure 1), at a level of quality not achieved by prior methods.

Table 2: **Input-view consistency.** CUPID achieves superior input view consistency, producing accurate pose, geometry and appearance alignment.

| Dataset | | Toys4K | | | GSO | | |
|---|---|---|---|---|---|---|---|
| Method | Pose | PSNR↑ | SSIM↑ | LPIPS↓ | PSNR↑ | SSIM↑ | LPIPS↓ |
| LaRa | × | 22.00 | 93.42 | 0.0884 | 19.81 | 91.61 | 0.1119 |
| OpenLRM | × | 26.41 | 80.17 | 0.1156 | 25.79 | 78.80 | 0.1268 |
| OnePoseGen | ✓ | 17.43 | 89.37 | 0.1174 | 14.87 | 86.46 | 0.1386 |
| Ours | ✓ | **30.05** | **96.81** | **0.0251** | **28.68** | **95.49** | **0.0354** |

Table 3: Comparison on CLIP image scores of novel views.

| Method | ViT-B/16 | ViT-L/14 |
|---|---|---|
| OnePoseGen | 0.7933 | 0.7193 |
| LaRa | 0.8334 | 0.7682 |
| OpenLRM | 0.8939 | 0.8410 |
| TRELLIS | 0.9465 | 0.9210 |
| Ours | **0.9501** | **0.9291** |

Table 4: Ablation studies of pose-aligned conditioning.

| Method | GT Geo & Pose | | | Sampled Geo & Pose | | |
|---|---|---|---|---|---|---|
| | PSNR | SSIM | LPIPS | PSNR | SSIM | LPIPS |
| (a) Baseline (TRELLIS) | 31.84 | 97.50 | 0.0219 | 27.47 | 95.64 | 0.0327 |
| (b) Position Embedding | 32.07 | 97.58 | 0.0211 | 27.56 | 95.67 | 0.0323 |
| (c) Latent (w/o Occ.) | 32.37 | 97.72 | 0.0201 | 27.85 | 95.87 | 0.0309 |
| (d) Latent (Occ.) | 32.39 | 97.77 | 0.0199 | 27.74 | 95.80 | 0.0313 |
| (e) Latent (Visual Feat.) | **34.86** | **98.24** | **0.0168** | **30.05** | **96.81** | **0.0251** |

Figure 5: **Qualitative comparison of various pose-aligned conditioning.** our method (e) achieves the best visual quality in terms of color fidelity and detail.

## 4.2 ABLATION STUDIES

We compare five pose-aligned conditioning variants: (a) the TRELLIS baseline; (b) adding DINOv2 positional embeddings to SLat latents; (c) concatenating DINOv2 feature volumes with view-conditioned voxel latents from the frozen SLat encoder; (d) as in (c) but zeroing latents of voxels occluded in the input view; and (e) concatenating additional visual features from a convolution layer applied to the input image before DINOv2 downsampling. From Table 4, method (c) and (d) validate the benefit of pose-aligned conditioning in generating geometry and appearance; our full method (e) provides the best visual quality by supplementing DINOv2 tokens with missing low-level cues, yielding better color and detail alignment, as shown in Figure 5. These gains persist with sampled geometry and pose, indicating our pose-aligned feature injection is robust to small variations.

## 5 RELATED WORK

**3D reconstruction from many images.** Complete 3D object reconstruction traditionally requires multiple views and Structure-from-Motion (Schönberger & Frahm, 2016; Schönberger et al., 2016). DUSt3R (Wang et al., 2024) predicts pixel-aligned point maps for image pairs, enabling efficient recovery of view-centric camera poses and partial geometry; subsequent works improve efficiency and flexibility (Wang et al., 2025a;d;b; Zhang et al., 2025; Yang et al., 2025a; Karaev et al., 2024; Yang et al., 2024; Liu et al., 2024d; Cabon et al., 2025). CUT3R processes images recurrently (Wang et al., 2025b); VGGT jointly predicts depth, point maps, and poses (Wang et al., 2025a); and MoGe introduces affine-invariant point maps for monocular geometry (Wang et al., 2025c). Unlike 2D-to-3D regression, our method generates 2D coordinates from 3D volumes, allowing for complete recovery of 3D objects and object-centric camera poses.

**3D reconstruction from one or a few images.** High-quality multiview data are hard to obtain in practice, motivating single- or sparse-view 3D reconstruction with large models (Xu et al., 2024a; Wei et al., 2024; Liu et al., 2024b; Tochilkin et al., 2024; Tang et al., 2024; Yinghao et al., 2024; Xu

Figure 6: **Multi-view conditioning.** When multiple input views are available, we fuse the shared object latent across flow paths, enabling camera, geometry and texture refinement across all images. Top: inputs; Middle: reconstructed 3D object and camera poses; Bottom: rendered images and geometry.

et al., 2024c). LRM (Hong et al., 2023) shows that complete 3D can be regressed from a single image under a fixed camera. PF-LRM (Wang et al., 2023) enables pose-free sparse-view reconstruction via differentiable PnP (Chen et al., 2022). Subsequent work improves single-view quality by augmenting novel views from 2D generative priors and using more efficient representations (Xu et al., 2024b; Yinghao et al., 2024; Liu et al., 2023a) or architectures (Chen et al., 2024). However, all methods are view-centric, not object-centric. Furthermore, these methods cannot generate diverse and plausible 3D objects that accurately explain the input image.

**3D reconstruction from generative priors.** Dreamfusion (Poole et al., 2022) pioneer 3D generation from 2D diffusion models (Rombach et al., 2021; Saharia et al., 2022). Following work Zero-1-to-3 (Liu et al., 2023b) fine-tune image diffusion models for novel view synthesis, enabling single image-to-3D generation and inspiring follow-ups (Shi et al., 2023; Watson et al., 2022; Li et al., 2024; Liu et al., 2024c; Gao et al., 2024; Höllein et al., 2024; Wu et al., 2024). Recent methods (Zhou et al., 2025; Ren et al., 2025; Song et al., 2025) utilize video diffusion priors to achieve greater view consistency. Yet, all these methods, which leverage 2D priors, require multi-view reconstruction to obtain the 3D object, sharing similar limitations of *many images to 3D reconstruction*. The very recent attempts (Huang et al., 2025; Yao et al., 2025; Dong et al., 2025) incorporate 3D generative priors (Xiang et al., 2025; Zhang et al., 2024b) for scene reconstruction, but relies on costly ad-hoc optimization (Yao et al., 2025) or extra scene datasets (Huang et al., 2025).

## 6 Discussion

We have presented CUPID, a pose-grounded generative framework that addresses single-image 3D reconstruction from a new perspective. By explicitly anchoring generation to a canonical 3D object representation, CUPID naturally integrates 3D priors learned in the canonical space with image inputs and enforces both geometric and photometric fidelity in generation via camera pose-aligned image conditioning. This simple but powerful grounding jointly enables high-fidelity single-view reconstruction and scene composition, significantly eases the acquisition of 3D assets.

Despite its strong performance and consistent gains over prior methods, CUPID has limitations. First, like most existing 3D object generation methods, it relies on an accurate object mask, and boundary errors in real images can degrade reconstruction quality. Second, lighting can be partially baked into textures. Further improvements require better material–light disentanglement. Third, our synthetic training images are mostly centered, making off-centered objects more challenging in real scene generation. Nevertheless, these are not fundamental limitations of our method and can be alleviated with better training data and supervision.

Looking forward, this generation-based reconstruction framework opens promising test-time extensions, particularly for multi-view scenarios. By leveraging multiple 2D images, our method can refine 3D reconstructions to align with all observations by fusing shared occupancy volumes during sampling, similar to Multi-Diffusion (Bar-Tal et al., 2023). As shown in Figure 6, this yields an SfM-like system, though challenges like conflicting poses from diverse views require advanced fusion schemes in future work. Furthermore, CUPID's joint modeling of 3D structure and pose enables bidirectional capabilities, such as flexible generation with known poses, or pose estimation from images of given 3D objects, facilitating tasks in virtual and augmented reality and embodied AI for robotics.

ETHICS STATEMENT

Our work advances 3D reconstruction for embodied AI, robotics, and augmented reality. However, it also poses ethical risks if misused. A primary concern is that bad actors might generate unauthorized 3D assets from 2D images, which could lead to intellectual property infringement, unauthorized replication of cultural artifacts, or privacy violations, such as non-consensual modeling.

To address these issues, our open-source code includes ethical guidelines. Additionally, we recognize that training datasets may contain biases, such as cultural underrepresentation; future efforts should prioritize inclusive data collection. We are also aware of the environmental impact of energy use during training and provide transparency regarding this matter. We are committed to evaluating societal impacts and promoting responsible usage of our technology to maximize its benefits.

REPRODUCIBILITY STATEMENT

The datasets used, including ABO (Collins et al., 2022), HSSD (Khanna et al., 2024), 3D-FUTURE (Fu et al., 2021), and Objaverse-XL (Deitke et al., 2023), are publicly available. TREL-LIS, upon which our work is built, is also publicly accessible via its official GitHub repository (Xiang et al., 2025). The Blender script for synthesizing 2D images can be adapted from the TRELLIS code base. We provide detailed training information in Appendix A.1. Our occlusion-aware model extends Amodal3R (Wu et al., 2025), which is publicly available; our customized modifications are detailed in Appendix A.2. While our least-squares pose solver is implemented in CUDA, an equally effective alternative can be found using the `calibrateCamera` function in OpenCV. Finally, we will open-sourced all our training and inference code to facilitate reproducibility.

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

# A APPENDIX

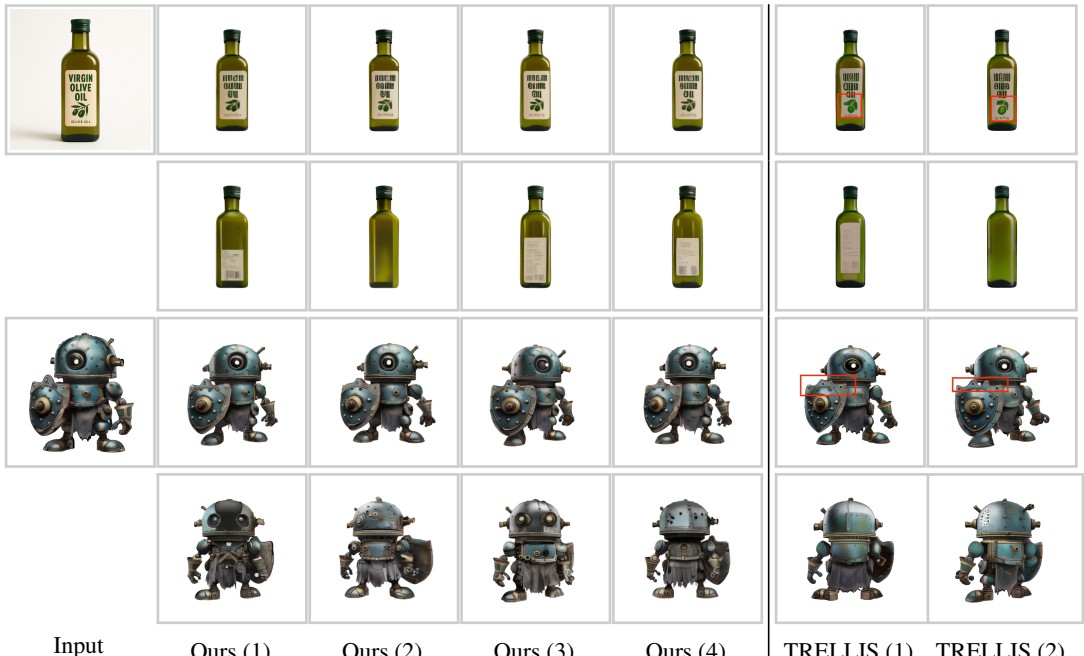

| Input | Ours (1) | Ours (2) | Ours (3) | Ours (4) | TRELLIS (1) | TRELLIS (2) |

Figure 7: **Diversity of our *Generative Reconstruction*.** We visualize canonical front and back views of generated 3D objects using random seeds (1-4). Given a single input image, our model synthesizes diverse hypotheses for unobserved regions while remaining highly consistent with visible regions. In contrast, the base 3D generator (Xiang et al., 2025) struggles to conform to the input image, or produce less diverse unobserved regions.

## A.1 IMPLEMENTATION

We train our model using several datasets, including ABO (Collins et al., 2022), HSSD (Khanna et al., 2024), 3D-FUTURE (Fu et al., 2021), and a subset of Objaverse-XL (Deitke et al., 2023), totaling approximately 260K 3D assets. These artist-curated datasets are predominantly aligned to a canonical frame where the ground plane corresponds to $z = 0$. Following TRELLIS (Xiang et al., 2025), we encode each asset into occupancy grids and structured latents that are suitable for training the flow transformer. The structured latents can be decoded into high-quality triangle meshes or Gaussian splats using the SLat decoders.

To enable pose-grounded generation, we render 24 conditioning images from random viewpoints for each asset, with augmented focal lengths ranging from 24 mm to 200 mm. We then convert the camera pose and the coarse structure into a 3D occupancy grid and corresponding UV volume $\{\mathbf{x}_i, \mathbf{u}_i(\boldsymbol{\theta})\}_{i=1}^{L}$, as described in the main paper. We first train a 3D VAE to encode the full UV volume of shape $64 \times 64 \times 64$ into 3D latents of shape $16 \times 16 \times 16$, which are concatenated with occupancy latents for training the first flow model $\mathcal{G}_{\mathbf{S}}$. Then, a second flow model $\mathcal{G}_{\mathbf{L}}$ is trained to generate the structured latents $\{\mathbf{f}_i\}_{i=1}^{L}$ based on the occupancy grid.

We initialize our models using the pretrained weights of TRELLIS. During training, we apply classifier-free guidance (Ho & Salimans, 2022) (CFG) with a drop rate of 0.1. Both $\mathcal{G}_{\mathbf{S}}$ and $\mathcal{G}_{\mathbf{L}}$ are trained using AdamW (Loshchilov & Hutter, 2017) at a fixed learning rate of $1 \times 10^{-4}$ for 500k and 100k steps, respectively. Training completes in approximately one week on 32 GPUs. At inference time, we use 25 sampling steps with classifier-free guidance strengths of 7.5 and 3.0 for $\mathcal{G}_{\mathbf{S}}$ and $\mathcal{G}_{\mathbf{L}}$, respectively.

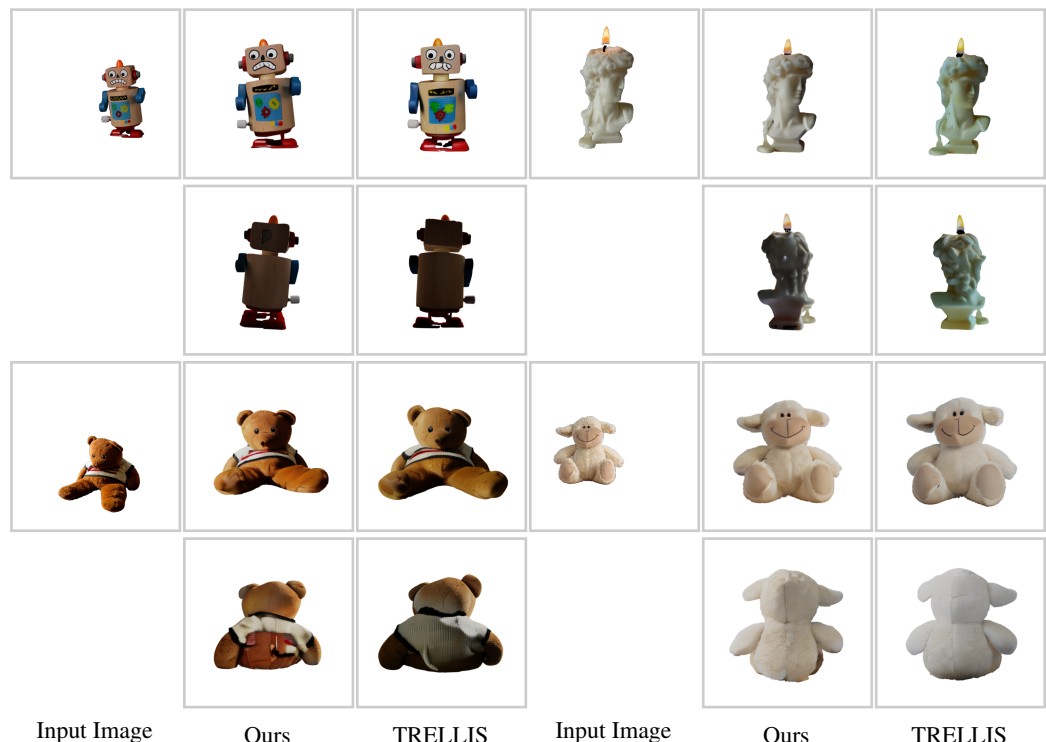

Figure 8: **Qualitative comparison:** *Generative Reconstruction* **vs.** *3D Generation.* Unlike standard 3D generation, which aims to create novel objects from images, our generative reconstruction is specifically designed to accurately recreate a particular object that replicates the visible regions of the input image while maintaining diversity in the invisible regions. This difference in objectives is crucial, as prior 3D generators (Xiang et al., 2025) are not optimized for this task and often produce artifacts such as *color drift* and *texture inconsistencies*. As demonstrated in the canonical front and back views, our method's pose-aligned image conditioning effectively reduces these issues, resulting in a reconstruction that remains true to the input.

## A.2 OCCLUSION-AWARE CONDITIONING.

To handle occlusion in complex scene reconstruction, our model leverages partial 3D object observations as conditions to generate complete objects. Our model takes a 2D occlusion mask $\mathbf{M}^{occ} \in \{0,1\}^{H \times W}$ as input alongside the visible object observation $\mathbf{I}^{cond}$. The mask $\mathbf{M}^{occ}$ identifies pixels that may belong to the object if occluders were removed, with values set to zero when no occlusion is present. Together, $\mathbf{M}^{occ}$ and the alpha channel of $\mathbf{I}^{cond}$ identify three pixel classes: (a) directly observed object pixels, (b) background pixels that must not contain the object, and (c) occluded pixels that may or may not contain the object.

We apply two modifications to both flow transformers to incorporate the mask. First, inspired by Amodal3R (Wu et al., 2025), we modulate the attention weight matrix during global condition injection via cross-attention using the mask. Specifically, the attention weight for each input token is computed by patching the mask to match the DINOv2 tokens and calculating the ratio of unmasked pixels in each patch. The logarithm of the weight values is added to the attention logits before applying the softmax operation. Second, for the geometry and appearance flow model that takes additional pose-aligned features, we concatenate the input image with the mask as an additional channel before feeding it into the convolution layer in the second stage.

During training, we randomly generate occlusion masks $\mathbf{M}^{occ}$ following Amodal3R and zero out the corresponding regions in $\mathbf{I}^{cond}$, preventing information leakage on occlusion regions and encouraging the model to reconstruct complete 3D objects from partial observations. At inference time, occlusion masks can be obtained heuristically or manually for scene reconstruction.

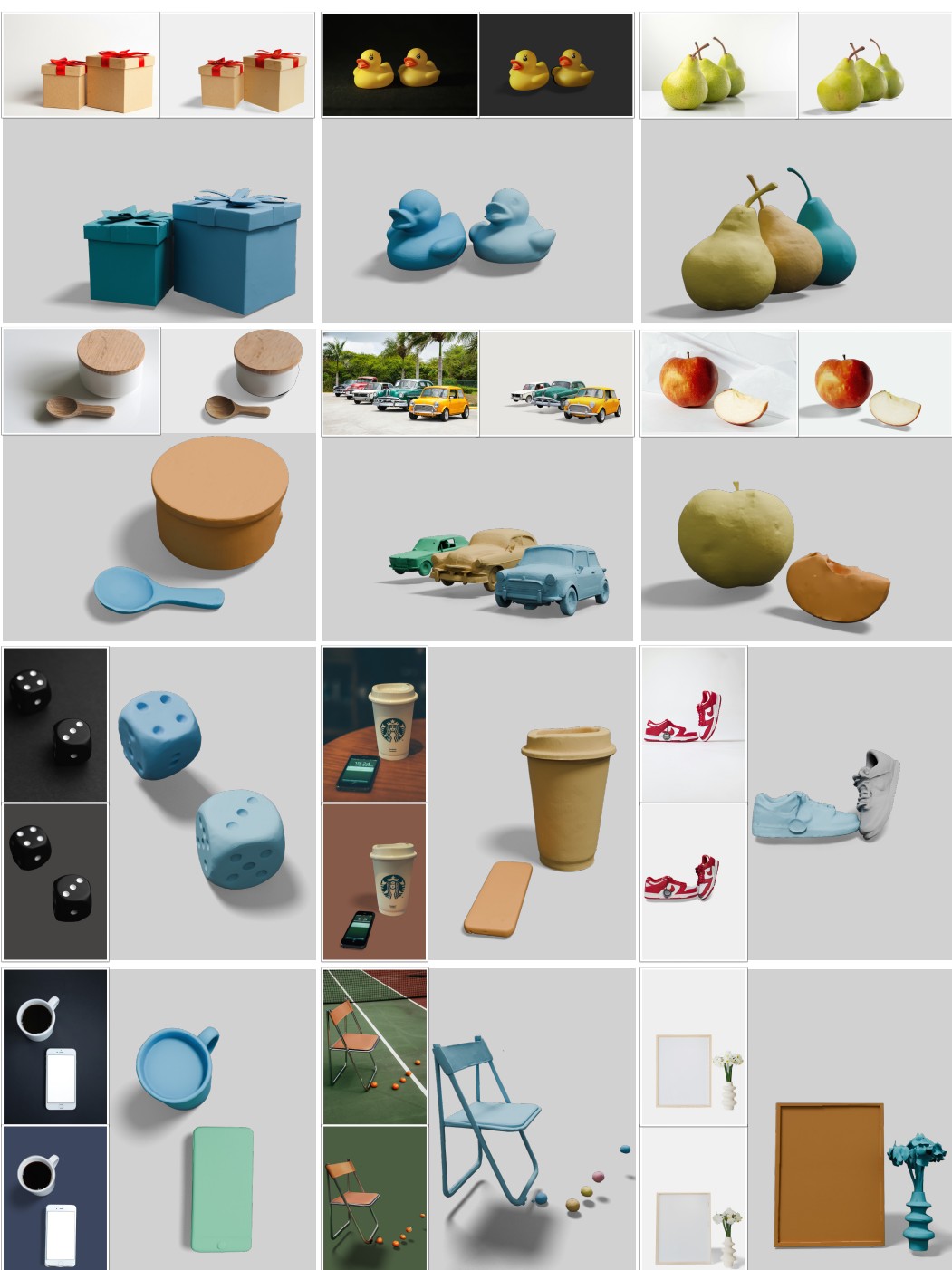

Figure 9: Additional examples of component-aligned scene reconstruction. For each example shown, the panels display: (top left) the input image, (top right or bottom left) the final rendered output, and (bottom) the reconstructed individual components, color-coded for clarity.

## A.3 GENERATIVE RECONSTRUCTION DIVERSITY.

Reconstruction methods like LRM (Hong et al., 2023) typically generate a single 3D object from one image. However, our approach not only surpasses LRM in terms of reconstruction quality but also produces multiple plausible interpretations—what we refer to as generative reconstruction. Since standard metrics do not effectively capture this diversity, we offer qualitative comparisons in Fig-

ure 7. The results demonstrate that our framework creates diverse 3D models with visible regions that consistently align with the input image. In contrast, conventional 3D generation methods Xiang et al. (2025) often struggle to maintain consistency in the visible regions of the objects they create. Furthermore, while directly processing an image with multiple objects may still yield a scene, the generation quality is often degraded as such inputs are out-of-distribution. In contrast, our method allows for sequential, component-aligned compositional reconstruction, as demonstrated by the additional examples in Figure 9.

### A.4 QUALITATIVE COMPARISON WITH 3D GENERATIVE METHODS

Since TRELLIS (Xiang et al., 2025) does not provide an explicit object-centric camera pose for the input image and post-hoc alignment is unreliable, we cannot conduct fair quantitative evaluation of reprojection consistency. We therefore focus on qualitative comparisons in Figure 8. For both methods, we render canonical object from front and back views. As shown, TRELLIS struggles to maintain texture consistency with the input image, exhibiting notable color drifting on the candle statue, teddy bear, and the lamb. In contrast, our method closely preserves the appearance of input. Importantly, although we incorporate pixel-level cues via back-projection, the model does not simply copy pixels: the generated back views remain coherent with the front, indicating learned view-consistent geometry and appearance. These results demonstrate that our mechanism effectively mitigates color drifting and texture inconsistency. We hypothesize that this limitation is common among 3D generators that lack localized, pixel-attended conditioning, including follow-up works of TRELLIS. We hope these findings can inspire future designs of 3D generators.

## B USE OF LARGE LANGUAGE MODELS (LLMS)

We used LLMs to assist with writing. The authors drafted the manuscript, and LLMs were employed for grammar correction, copy-editing, and improving clarity and fluency. No novel scientific ideas, analyses, or experimental results were produced by LLMs. All LLM-assisted text was reviewed, revised, and approved by the authors, who take full responsibility for the content.

