# OpenReview forum: "CUPID: Pose-Grounded Generative 3D Reconstruction from a Single Image"
_ICLR.cc/2026/Conference — ICLR 2026 Conference Withdrawn Submission_

### Official Review · Reviewer_qbaU · 2025-10-28

**Soundness:** 2
**Presentation:** 2
**Contribution:** 2
**Rating:** 2
**Confidence:** 3

**Summary:**

This paper proposes CUPID, a two-stage image-to-3D generation pipeline based on a cascaded flow model. In the first stage, the method generates an occupancy cube and a UV cube conditioned on the input image. The occupancy cube encodes the 3D spatial locations of objects, while the UV cube establishes 2D-to-3D correspondences, from which an object-centric camera pose can be inferred via a PnP solver. The second stage employs a Geometry and Appearance decoder to produce 3D representations (either 3D Gaussians or meshes) based on pixel-aligned features.

The experiments (qualitatively and quantitatively) show that the proposed method generates 3D assets that are more faithful to the input images and achieve higher quality for novel views.

**Strengths:**

1. The proposed method predicts more accurate visible geometry than prior image-to-3D approaches (e.g., LRM and LaRa), and significantly outperforms them in terms of input-view consistency.
2. The method also achieves superior novel-view quality compared to existing baselines.
3. The proposed pipeline is versatile and flexible: it supports generating multiple objects in the image and even the ones that are occluded by others.

**Weaknesses:**

1. I am not fully convinced that predicting canonical object-centric camera poses is necessary. In fact, the concept of an “object-centric” camera pose is somewhat ill-defined, raising the question of what even a canonical object-centric frame is (which is directly correlated with the training data distribution). The pipeline could be simplified by performing all generations in the camera-centric space, where the camera pose remains fixed and shared across objects. In this formulation, the PnP solver may no longer be needed. To verify this, the authors should compare against a camera-centric baseline in which all components (e.g., Occupancy and UV cubes) are predicted in a fixed camera space.
2. If the proposed canonical object-centric formulation offers only marginal improvements, the authors should clarify how their method fundamentally differs from TRELLIS. As it stands, the proposed approach appears to be an incremental extension of TRELLIS.
3. Tables 1 and 2 should include TRELLIS as a baseline for fair comparison (as currently, baselines are all regression-based methods).
4. This concern is related to (1) and (2). In L322–323, the paper states that CUPID outperforms LRM and LaRa on monocular geometry accuracy, suggesting that it can better recover object geometry. However, these results alone do not isolate the contribution of the canonical object-centric reconstruction. The improvement could instead stem from CUPID’s use of generative modeling (and pretrained priors from TRELLIS), which is typically more robust to uncertainty in single-view reconstruction than regression-based methods like LRM and LaRa.
5. Figure 4 should also visualize novel views of the generated 3D outputs, rather than showing only the input view.

**Questions:**

1. It is unclear whether Table 4 reports only input-view consistency metrics. The authors should also evaluate and report performance on novel views, if applicable. And what's the difference between GT Geo & Pose and Sampled Geo & Pose?

---

### Official Review · Reviewer_nFtK · 2025-10-30

**Soundness:** 3
**Presentation:** 3
**Contribution:** 3
**Rating:** 6
**Confidence:** 5

**Summary:**

This paper introduces CUPID, a novel pose-grounded generative framework that bridges the gap between 3D reconstruction and generation by jointly estimating camera pose, 3D shape, and texture from a single image. The method innovatively represents camera pose as dense 3D-2D correspondences within a voxel grid (termed "UV cube"), enabling robust pose recovery via PnP solvers. Through a two-stage cascaded flow matching pipeline, CUPID first generates coarse geometry and pose, then refines appearance using pose-aligned local image features. The approach demonstrates strong performance in both object-level reconstruction and scene-level compositional generation, outperforming existing methods in reconstruction fidelity and achieving competitive pose estimation accuracy.

**Strengths:**

1.Novel Pose Representation: The proposed dense 3D-2D correspondence representation for camera pose is theoretically grounded and practically effective. This representation naturally integrates with the 3D generation process and enables more robust pose estimation compared to low-dimensional parameterizations.

2.Effective Feature Conditioning: The pose-aligned feature injection mechanism, which combines both high-level semantic features (DINOv2) and low-level visual cues, demonstrates substantial improvements in geometric accuracy and appearance fidelity.

3.Scalable to Scene Reconstruction: The method elegantly extends to multi-object scene reconstruction through occlusion-aware generation and component-aligned composition, showing practical applicability beyond single-object scenarios.

4.Comprehensive Evaluation: The paper provides thorough experimental validation across multiple datasets and tasks, demonstrating state-of-the-art performance in both reconstruction quality and pose estimation.

**Weaknesses:**

1.Limited Comparison with 3D Generative Baselines: While the paper extensively compares with reconstruction-oriented methods (e.g., LRM, LaRa) and pose estimation methods, it provides only qualitative comparisons on CLIP score with TRELLIS. A more comprehensive quantitative evaluation against contemporary 3D generation approaches (e.g., TRELLIS, Hunyuan3D) would strengthen the claims of superiority.

2.Insufficient Pose Estimation Analysis: Given that the entire refinement stage depends on accurate pose estimation from the first stage, a more detailed evaluation of pose estimation accuracy—including quantitative comparisons with specialized pose estimation methods and analysis of failure cases—would be valuable.

**Questions:**

I find the approach of having the model directly predict the UV cube to be highly ingenious and innovative. If this method is proven to outperform both the paradigm of predicting RT directly from learnable tokens and existing approaches similar to "camera as ray[1]" it would represent a significant breakthrough. This would free the construction of UV-XYZ in the Perspective-n-Point (PnP) framework from being limited to rendered images, thereby enabling better prior usage in 3D generative models. However, this paper lacks sufficient validation regarding the performance of this paradigm in pose prediction. Consequently, I tend to recommend a Borderline Accept (BA) decision. If the authors can supplement this experimental validation, I believe this paper deserves acceptance as a high-quality work.

[1] Cameras as Rays: Pose Estimation via Ray Diffusion （ICLR2024）

---

### Official Review · Reviewer_iTpF · 2025-10-31

**Soundness:** 3
**Presentation:** 2
**Contribution:** 2
**Rating:** 2
**Confidence:** 4

**Summary:**

This paper proposes a generation-based 3D reconstruction method named CUPID, which jointly estimates the object-centric camera pose, recovers canonical 3D geometry, and reconstructs texture from a single RGB image. CUPID formulates 3D reconstruction as a conditional sampling process from a learned distribution of 3D objects, and jointly generates voxels and pixel-voxel correspondences via a two-stage flow matching pipeline. Its core goal is to address the limitations of traditional reconstruction (e.g., occluded region missing) and generative models (e.g., poor view consistency). The paper validates the method’s effectiveness by comparing it with representative baselines across three key metrics: monocular geometry accuracy, input-view consistency, and novel-view consistency.

**Strengths:**

The paper is well structured and the writing is easy to understand. The proposed CUPID outperforms leading baselines across multiple critical evaluation dimensions, demonstrating its comprehensive superiority.

**Weaknesses:**

1) The core design of the paper lies in "binding explicit object-centric pose with 3D generation," but it fails to fully demonstrate the unique value of this design compared to implicit pose utilization in generative models. Existing 3D generative models (e.g., TRELLIS) can implicitly learn view correlations through multi-view features without explicit pose modeling. While the paper shows performance improvements over TRELLIS, it does not decompose the supporting role of "object-centric pose" in downstream tasks (e.g., scene composition, robotic grasping). For instance, the paper’s Figure 1 and Figure 3 demonstrate scene-level compositional reconstruction capabilities, but it does not quantify how "object-centric pose" (anchored to a canonical model) outperforms "view-centric implicit pose" (with view-dependent coordinate systems) in such tasks. This makes the argument for innovativeness insufficient.

2) Insufficient Experimental Support for Core Modules:
The UV Cube Decoder is the core module for explicit pose modeling, yet the paper does not report key pose estimation metrics (e.g., Relative Rotation Error (RRE), Relative Translation Error (RTE)) to verify its accuracy. It also lacks comparisons with state-of-the-art monocular pose estimation methods, making it impossible to confirm whether the estimated pose is sufficiently reliable to support subsequent 3D generation optimization.
The ablation study (Table 4) only verifies the effect of "pose-aligned feature injection" but does not design a control experiment to isolate the impact of explicit pose (e.g., removing the UV Cube Decoder and relying solely on TRELLIS’ implicit pose). This makes it impossible to distinguish the independent contributions of "pose estimation" and "feature injection" to 3D generation quality, leaving the gain of explicit pose unproven.

3) Incomplete Ablation and Comparative Experiments:
Point-map regression methods (VGGT, MoGe) only participate in the evaluation of monocular geometry accuracy and are excluded from input-view consistency and novel-view consistency tests, with no explanation provided. Moreover, view-centric 3D reconstruction methods (LRM, LaRa) are relatively outdated (published in 2023 and 2024, respectively), and no comparisons with recent methods (e.g., ReconViaGen 2025, NOVA3R 2025) are included. Furthermore, the paper fails to explain why VGGT outperforms CUPID in monocular geometry metrics.

**Questions:**

1) The core design of the paper lies in "binding explicit object-centric pose with 3D generation," but it fails to fully demonstrate the unique value of this design compared to implicit pose utilization in generative models. Existing 3D generative models (e.g., TRELLIS) can implicitly learn view correlations through multi-view features without explicit pose modeling. While the paper shows performance improvements over TRELLIS, it does not decompose the supporting role of "object-centric pose" in downstream tasks (e.g., scene composition, robotic grasping). For instance, the paper’s Figure 1 and Figure 3 demonstrate scene-level compositional reconstruction capabilities, but it does not quantify how "object-centric pose" (anchored to a canonical model) outperforms "view-centric implicit pose" (with view-dependent coordinate systems) in such tasks. This makes the argument for innovativeness insufficient.

2) Insufficient Experimental Support for Core Modules:
The UV Cube Decoder is the core module for explicit pose modeling, yet the paper does not report key pose estimation metrics (e.g., Relative Rotation Error (RRE), Relative Translation Error (RTE)) to verify its accuracy. It also lacks comparisons with state-of-the-art monocular pose estimation methods, making it impossible to confirm whether the estimated pose is sufficiently reliable to support subsequent 3D generation optimization.
The ablation study (Table 4) only verifies the effect of "pose-aligned feature injection" but does not design a control experiment to isolate the impact of explicit pose (e.g., removing the UV Cube Decoder and relying solely on TRELLIS’ implicit pose). This makes it impossible to distinguish the independent contributions of "pose estimation" and "feature injection" to 3D generation quality, leaving the gain of explicit pose unproven.

3) Incomplete Ablation and Comparative Experiments:
Point-map regression methods (VGGT, MoGe) only participate in the evaluation of monocular geometry accuracy and are excluded from input-view consistency and novel-view consistency tests, with no explanation provided. Moreover, view-centric 3D reconstruction methods (LRM, LaRa) are relatively outdated (published in 2023 and 2024, respectively), and no comparisons with recent methods (e.g., ReconViaGen 2025, NOVA3R 2025) are included. Furthermore, the paper fails to explain why VGGT outperforms CUPID in monocular geometry metrics.

---

### Official Review · Reviewer_vvak · 2025-11-03

**Soundness:** 2
**Presentation:** 4
**Contribution:** 3
**Rating:** 4
**Confidence:** 5

**Summary:**

CUPID tackles single-image pose-grounded 3D reconstruction, estimating an object-centric camera pose and producing a canonical 3D object with geometry and texture. It introduces a joint generative formulation over voxels and dense 3D–2D correspondences in a shared latent space. A two-stage flow pipeline first samples an occupancy and UV cube to recover pose via PnP, then refines geometry and appearance using pose-aligned image features. Experiments on Toys4K and GSO show >3 dB PSNR gains and >10% Chamfer Distance reduction over baselines while matching monocular pose estimators on accuracy. Overall, CUPID yields pose-accurate, high-fidelity single-view 3D and supporting component-aligned scene composition.

**Strengths:**

- Clearly formulates an object-centric, canonical pose setting, and learn it from dense 3D–2D correspondences.
- Superior input-view color faithfulness over TRELLIS.
- Sufficient experiments on the input view and reasonable ablations.
- It further supports component-aligned scene reconstruction.

**Weaknesses:**

- The “first to jointly model 3D objects and dense 2D–3D correspondences” claim feels overstated; prior works such as PF-LRM and SpaRP already couple geometry with correspondences/pose in generation/reconstruction. Relying on a distant first-page footnote to narrow “object” to the canonical frame risks being misleading, as the main novelty is better framed as the canonical, object-centric formulation.
- The paper lacks canonical-specific diagnostics that would substantiate the claimed advantage. Suggested additions include: variance of recovered azimuth under input viewpoint perturbations, cross-instance alignment error in the canonical frame, symmetry-aware pose consistency, and ablations isolating the role of dense correspondences in enforcing canonical orientation.
- Empirical focus is largely on the single input view; broader geometry evaluation is limited. A fair, full-geometry quantitative comparison to TRELLIS (and others) could be done by registering predictions into a common frame and reporting mesh/point-cloud metrics beyond the front view. From qualitative comparisons (Fig. 8 - robot), geometry sometimes appears less faithful than TRELLIS.

**Questions:**

I’d be happy to raise the rating if you provide more concrete evidence of canonical understanding.

- In the teaser, why not visualize the canonical space (e.g., show the object in the canonical cube and the recovered camera pose)?
- Is Objaverse-OA (Orientation matters), the canonically aligned subset of Objaverse, used for training in this work?

---

### Note · Authors · 2025-11-13

**Comment:**

We thank you for your time and constructive feedback.

After careful review, we identified a core misunderstanding of our work's primary goal:

1. A misperception of our task definition, which jointly reconstructs canonically posed objects and object-centric cameras (Reviewer qbaU questions the need to decouple these, while Reviewer vvak expresses reservations about canonicalness definition).
2. A misinterpretation of our technical contribution, which bridges 3D generation and reconstruction through explicit pose estimation (Reviewers vvak and qbaU request comparisons with 3D generation models, while Reviewer iTpF seeks comparisons with concurrent, non-public works).

Although we are withdrawing the paper to enable substantial revisions, we wish to clarify these points for the record and highlight the specific research gap we address.

---

## Our Formulation: Joint Generation of Canonically-posed Object and Camera Pose

Our method disentangles **intrinsic** object properties (shape and texture) from **extrinsic** observation properties (camera pose). This separation is critical for applications like robotic manipulation, where success depends on intrinsic geometry rather than viewpoint.

Regarding canonical orientation (Reviewer vvak): our training data are artist-curated assets in display-ready frames, naturally encoding human priors on canonicalness. While manually labeled small subsets (e.g., Objaverse-OA, Lu et. al., 2025 [1]) could be used, we find them unnecessary. True canonical frames are often ambiguous, and individuals may perceive different orientations (e.g., left- vs. right-handed users with a cup). Our learned representation captures the dominant mode from the data itself without explicit labeling.

> [1] Orientation Matters: Making 3D Generative Models Orientation-Aligned. Yichong Lu, Yuzhuo Tian, Zijin Jiang, Yikun Zhao, Yuanbo Yang, Hao Ouyang, Haoji Hu, Huimin Yu, Yujun Shen, Yiyi Liao. NeurIPS 2025.

Below we address specific techniques concerns.

**Q1: What is the benefit of generating using an object-centric frame instead of directly in camera space? (Reviewer iTpF, qbaU)**

**A:** There are three advantages in the paper:

1. **Camera Intrinsic Estimation:** Unlike view-centric approaches that assume **fixed camera intrinsics (K)**, our method can jointly infer the full camera projection including rotation (R), translation (T), and intrinsics (K). Even when aligning objects to camera orientation, a PnP solver is still necessary to recover intrinsic K.
2. **Multi-view Compatibility**: View-centric methods require N distinct 3D latents for N views. Our decoupled approach produces one shared intrinsic 3D latent with N extrinsic pose latents, enabling **test-time multi-view aggregation** without retraining.
3. **Simplified Learning:** Generating objects in a consistent canonical orientation disentangles object identity from viewing angle, making it easier to learn. Canonical reduces modality count from (N objects × N views) to (N objects).

**Q2: What is the benefits of estimating the camera pose for a generated 3D model? (Reviewer iTpF, qbaU)**

**A:** Camera pose is essential for bridging generation and reconstruction. Without it, we cannot: (i) pose-aligned condition generation on input images for high-fidelity results, (ii) ensure generated 3D content reproduces the input view, or (iii) enable applications like component-aligned reconstruction that require precise object-to-camera alignment.

---

## Core Contribution: Bridging 3D Generation and Reconstruction

We observed a crucial gap between two major lines of work in 3D vision:

- **3D Generation (e.g., TRELLIS):** Excels at generating 3D models from single images but doesn't model camera pose, so **it cannot recreate the input image without external pose estimators**.
- **3D Grounding and Reconstruction (e.g., VGGT):** Faithfully recovers visible 3D structure and camera pose deterministically but cannot generate plausible geometry for occluded regions.

**Our work addresses the gap between these two directions.** Our primary claim is that jointly estimating pose significantly improves the **fidelity (input view consistency)** of generated 3D objects, with applications in component-aligned and multi-view reconstruction. We do **NOT** claim to advance state-of-the-art **quality (aesthetics or geometric detail)** of 3D generators, as it is not our focus.

**Q3: Could the evaluation be extended to include a broader range of 3D generation methods? (Reviewer vvak, nFtK, qbaU)**

**A:** Our primary goal is to improve **input-view consistency** through explicit pose estimation, not advancing raw 3D generation quality. This focus is validated in **Tables 1 & 2**, **Figure 4**, and ablation studies, which show significant gains in input view fidelity. **Table 3** further confirms that our pose-aligned conditioning does **not** compromise novel-view quality (as measured by CLIP scores).

Since most 3D generation methods do not output camera pose, direct end-to-end comparison is not feasible **without adding external pose estimation as a post-processing step**. We therefore evaluate against **OnePoseGen** in our paper, a strong proxy that combines a SOTA generator (Higen3D/Trellis) with a leading 6DoF pose estimator (Foundation Pose).

Notably, our core contribution—**bridging generation and reconstruction via pose alignment**—is **framework-agnostic**. While demonstrated with TRELLIS, it can be applied to other primitive-based generators (e.g., Point-E, MosaicSDF). Given our targeted objective of enhancing view faithfulness, we prioritize comparisons that probe this specific benefit rather than competing on 3D shape quality that do not contribute to our claims.

**Q4: Is the pose reliable enough to support pose-aligned conditioning? What is the impact of explicit pose over “implicit pose” on 3D generation quality? (Reviewer iTpF)**

**A:** Table 4's ablation validates our estimated pose effectiveness. The PSNR improvement with our predicted pose is similar in magnitude to using GT pose and canonical geometry, indicating sufficient accuracy for effective pose-aligned conditioning. Moreover, result (a) directly compares with base TRELLIS without pose conditioning. Since TRELLIS doesn't explicitly model camera pose, there's no pixel-to-voxel correspondence to leverage, making it unclear how to meaningfully utilize any "implicit pose.”

---

## Other Specific Questions

**Q5: Why was there no comparison with point-map regression methods on input-view consistency and novel view synthesis? (Reviewer iTpF)**

**A:** Point-map methods predict pixel-aligned depth or point clouds, achieving perfect input-view consistency by directly sampling colors from the input image. However, they only predict geometry for visible surfaces and cannot generate plausible completions for occluded regions. Since novel-view synthesis is not their primary objective, **they address a fundamentally different task**, making direct comparison less meaningful.

**Q6: Are there more recent 3D reconstruction baselines that should be included (Reviewer iTpF)?**

**A:** We compared against relevant and established state-of-the-art methods. The suggested baselines (ReconViaGen [2][3], NOVA3R [4]) were **non-public**, **concurrent** submissions appearing **ONLY in ICLR 2026** at the time of our work (19 Sept 2025). Per academic standards, we cannot compare against unavailable papers :)

> [2] ReconViaGen: Towards Accurate Multi-view 3D Object Reconstruction via Generation. https://openreview.net/forum?id=z0QLeooEEf (*Submitted on* 19 Sept 2025)

> [3] ReconViaGen: Towards Accurate Multi-view 3D Object Reconstruction via Generation. https://arxiv.org/abs/2510.23306  (*Submitted on 27 Oct 2025*)

> [4] NOVA3R: Non-pixel-aligned Visual Transformer for Amodal 3D Reconstruction. https://openreview.net/forum?id=c0QRZMKwSb (*Submitted on* 19 Sept 2025)

**Q7: What is the pose metric for the UV decoder, and how does the prediction accuracy compare to state-of-the-art monocular pose estimators? (Reviewer iTpF)**

**A:** Our UV VAE achieves average RRE/RTE of 0.46/0.45 degrees and 95th percentile RRE/RTE of 1.14/1.14 for pose autoencoding. Unlike multi-view approaches (e.g., VGGT) that report pose accuracy relative to the first frame, single-view generation tasks cannot report relative pose accuracy metrics. Therefore, following MoGe, we report monocular geometry alignment metrics as an effective proxy for measuring pose accuracy.

**Q8: Would predicting R,T directly from learnable tokens or in a way similar to Camera as Ray be relevant alternatives to evaluate? (Reviewer nFtK)**

A: We thank the reviewer for this insightful suggestion. Directly regressing R,T from learnable tokens is an interesting direction. However, most existing 3D generators operate with 3D tokens, whereas the referenced baselines use 1D (camera vector) or 2D (ray map) tokens, requiring more architectural changes and potentially complicating learning. We will explore suitable adaptations and include this comparison in the revised version.

---

We believe our work addresses a timely and important gap in 3D vision. We have decided to withdraw the paper to substantially revise the manuscript. We thank the reviewers and AC again for their time and engagement.

Sincerely,

The Authors of Submission 2929

**Withdrawal Confirmation:**

I have read and agree with the venue's withdrawal policy on behalf of myself and my co-authors.